# Gateway to Outdoors: Partnership and Programming of Outdoor Education Centers in Urban Areas

**Paige O'Farrell [1] and Hung-Ling (Stella) Liu [2],*** 

[1]   City of Sioux City Parks and Recreation Department, Sioux City, IA 51102, USA; pofarrell@sioux-city.org
[2]   Deaprtment of Health and Nutritional Science, South Dakota State University, Brookings, SD 57007, USA
*   Correspondence: stella.liu@sdstate.edu; Tel.: +1-605-688-6163

**Abstract:** The purpose of the study is to understand the challenges and opportunities of urban outdoor education centers in partnership and programming. The context for this study involves efforts by all-season outdoor education centers, Outdoor Campus (OC), in two urban areas in South Dakota (SD). Outdoor education scope and social-ecological framework were applied to guide this qualitative study. Semi-structured interview questions were used to interview eight outdoor educators in 2019, including four individuals from each service location composed of three males and five female educators. Qualitative content analysis was applied to identify common themes and essential quotations that emerged from the data analyzed through the interviews. Three main themes emerged: (1) gateway to our outdoor legacy (2) working together for outdoor education, including three sub-themes: formal partnership, programmatic partnership, and finding balance in partnership, (3) challenges as opportunities in outdoor education programs, including two sub-themes: common challenges and evolving process.

**Keywords:** outdoor education; outdoor skills; partnership; outdoor programs; outdoor education in urban areas

---

## 1. Introduction

Outdoor recreation helps individuals of all ages gain physical activity, reduce stress, increase life satisfaction, and enhance interpersonal interaction [1–3]. Outdoor education is also used to increase public support of conservation endeavors and environmental literacy [4]. Both consumptive (e.g., fishing and hunting) and non-consumptive (e.g., hiking and kayaking) outdoor activities contribute to such conservation and restoration efforts through recreation-related spending, including equipment and license purchases. However, the declining outdoor recreation participation in the United States has been a concern for many agencies involved in managing public recreation areas [5]. The lack of diversity is also a concern in promoting outdoor recreation. Approximately 75% of white males participate in outdoor recreation, whereas females, younger generations, and ethnic minority groups showed a significantly lower rate of participation [6]. Therefore, it is essential to apply and implement proper and efficient programs and strategies to reach out to these groups, typically the most inactive outdoor recreation participants [7]. Outdoor education centers are used as gateways to engage with the public for promoting outdoor activities through educational efforts and outreach to serve people offering a wide range of interests and experiences in the outdoors. Especially in urban areas, outdoor education centers provide opportunities for recreational involvement and allow personal growth and learning in a unique setting [8] and allow individuals to connect and bond with their communities, with other individuals, and their environment through a variety of programs [9]. Considering a broader use of green infrastructure or ecosystem service in urban areas, outdoor education centers also could promote ecosystem health and human well-being in the community [10] and further the

understanding between urban green space and public health within the context of environmental justice [11]. However, due to the limited resources, many outdoor education centers and sustainable programs must rely on partnerships to leverage resources, such as facilities and equipment, staff, skills, and expertise, to accomplish shared goals [5].

South Dakota, from a cultural and historical perspective, provides a profound outdoor recreation opportunity for residents and visitors. It is especially famous for fishing and boating on the Missouri River and its reservoirs, and hunting culture and resources across the state. Although these activities are considered as outdoor recreation in the modern era, they were everyday-life, survival activities in the early days of South Dakotans [12]. Outdoor education is believed to be an essential effort to continue this outdoor legacy in South Dakota. However, some concerns have been raised about how outdoor heritage activities (e.g., fishing, hunting) might damage our natural environment and harm Indigenous' eco-social structure and communities [13], especially in a state, like SD with a high Native American population (9%) [14]. Such environmental conflicts and disparities of environmental health and justice of Native American communities have been on-going issues and unsolved problems in South Dakota [15]. However, it is worth noting that South Dakota residents in a current statewide study showed a strong awareness and recognition of needs with a desire to push for better protection, promotion, and enhancement of SD's heritage and Native American heritage about conservation efforts and outdoor recreation promotion [12]. A statewide education effort is an integral approach of the state wildlife management agency to provide sustainable outdoor recreational opportunities for the public [16]. With a growing population and decline in outdoor participation in urban areas, two all-season outdoor education centers were developed at the end of the 20th century for serving and promoting outdoor recreation in South Dakota urban areas for future generations.

The purpose of this study is to understand the challenges and opportunities of urban outdoor education centers in applying innovative programs to reach out to the public with a wide range of interests and experience in the outdoors and utilizing partnerships to create a social network in the community for enhancing the culture of outdoor recreation and environmental conservation. The context for this study involves efforts by all-season outdoor education centers of two urban areas in South Dakota (SD). Previous studies have identified partnership benefits and barriers to outdoor education. The current study builds on that knowledge by investigating how outdoor education centers act as a gateway entity serving as the center of building community support and networks for outdoor education, and promoting outdoor recreation in urban areas with a specific focus on partnerships and program innovations.

## 2. Literature Review

This qualitative study used two theoretical frameworks to feature the importance of programming and partnership of outdoor education centers in urban areas. The authors applied Higgins and Loynes' [17] outdoor education scope to explain the range of programs offered at the outdoor centers and to investigate the emphasis areas and changes of current programs. Additionally, the social–ecological framework was used to describe the dynamic relationship and partnership between the outdoor education centers and other related partners and to illustrate challenges and opportunities for outdoor recreation participation-related topics [18]. More specifically, partnership and program innovations were the two focus areas in the study for understanding how outdoor recreation centers leverage resources for serving the communities and how outdoor education-related programs lead to new directions for promoting the outdoors.

### 2.1. Education Programs in Outdoor Education Centers

Outdoor education centers around the nation play vital roles in interacting with and educating the public about outdoor knowledge, skills, and appreciation while supporting the overall vision and mission of their affiliated agencies and organizations. The study utilized the work of Higgins and Loynes particularly a conceptual model created to explain the scope of education programs provided

in urban outdoor education centers [17]. Three main domains are incorporated in direct experience of outdoor learning, including environmental education, outdoor activities, and personal and social development. Environmental education is commonly understood as the study of landscape, such as biology, geography, history, and culture. Outdoor activities incorporate skill acquisition related to those activities, such as kayaking, climbing, and fishing. Personal development in the model is used to promote qualities like self-esteem and self-awareness in people's lives, while social development is about interpersonal skills and working in groups. It is essential to focus on one or more areas and be sensitive to other opportunities that might guide such education experience within the complementary areas [17].

Special attention has been paid to school children and youth in outdoor education. Some short-term effects include increasing the comfort level in the outdoors, viewing humans as part of nature, and increasing preferences to visit parks and go outside rather than seeing a movie or playing video games indoors. Some long-term attitudes fade over time, but environmental knowledge, and environmental awareness remained [19]. Some studies focused on outdoor education at school settings in various learning opportunities, such as physical activity/education [20–23], science learning [21,24], environmental education [25], and a holistic learning experience [1,24]. Most importantly, outdoor education programs also teach children and youth beneficial life skills, such as communication, leadership, and problem-solving, that are transferable into other aspects of students' lives [26].

Other studies addressed the importance and impacts of outdoor education resources outside of school systems in the community available for broader audiences, including adults, children/youth, and family. Such services could be provided by local, state, and federal land management agencies [27], university extension/outreach offices [21], and private entities and organizations [28]. Among these service providers, public government agencies are commonly reviewed as the most promising entities serving multiple purposes for education, recreation, environmental protection, and health promotion to the public. A public outdoor education center in an urban area has been used as an informal approach to providing a place to learn and a process of experiential learning [29]. Green spaces (e.g., parks, greenways, open space, forests, gardens) are widely recognized as important contributors to health for residents in urban areas [10,30–32]. However, inequitable access to natural environments, especially in populated areas or urban communities, has been a significant concern of outdoor recreation professionals [8]. With a long history of budgetary limits and philosophical debates in public services for outdoor recreation, many outdoor education and recreation centers in the public sector have been facing critiques of appropriate services provided by governmental agencies and challenges related to charging or increasing fees for public outdoor education opportunities [33,34].

### 2.2. Social–Ecological Model and Partnership

The social–ecological framework was used to investigate the impact of public outdoor education centers in urban areas under a social and community context and understand how such outdoor education centers create a community support network through partnerships with other organizations in promoting outdoor recreation. This framework originated from Bronfenbrenner's ecological system model [35] to illustrate the interaction between individual and various social and environmental factors and how each level impacts individuals' development. In outdoor recreation specifically, Larson et al. [18] illustrated the interacting influence among different layers of social structure based on the scale of influence, including micro (e.g., individual, family), meso (e.g., interactional community), and macro (e.g., border society) levels (Figure 1). The social–ecological conceptual framework posits that individuals' outdoor recreation participation is influenced by higher-social order, environmental, and policy-related structures [18]. In the study, outdoor education centers in urban areas are reviewed as the epicenters for providing services to the public and working with diverse community groups and individuals at a meso level of social structure in the urban areas. These outdoor education centers also are the reflection of the state government's support, practice, and policies in promoting outdoors and overall state culture and atmosphere.

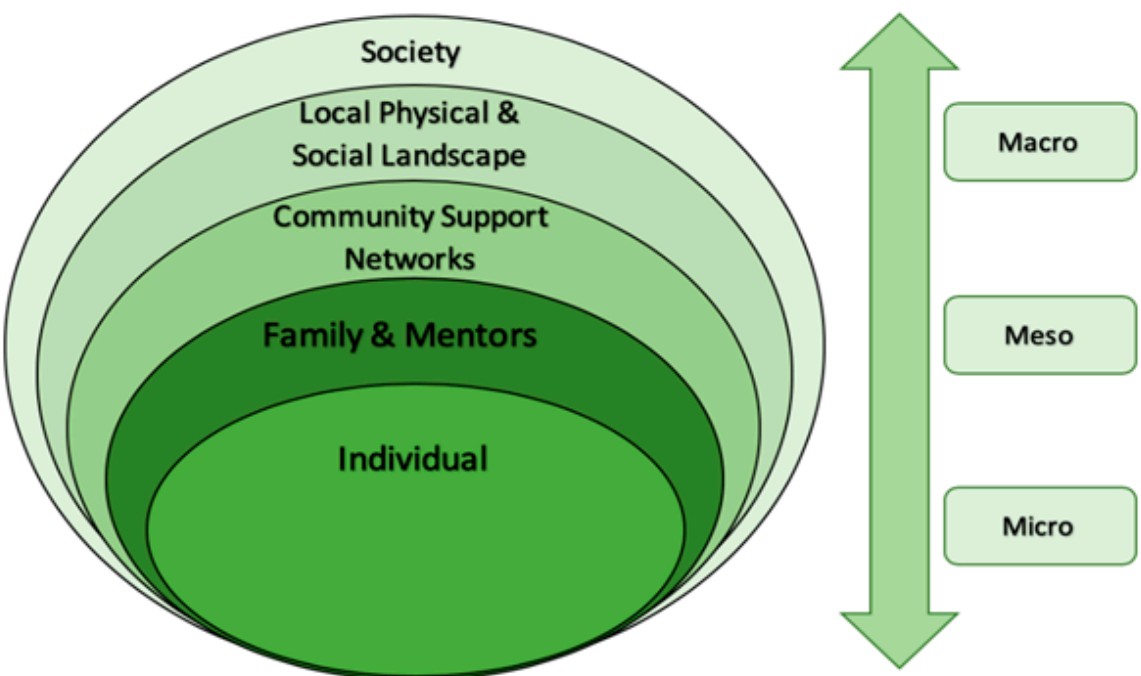

**Figure 1.** Nested levels of social structure for outdoor education (adapted from Larson et al. [18]).

Outdoor recreation and education heavily rely on effective partnerships to serve the urban population [5,36]. Partnerships allow organizations to accomplish their initiatives more quickly by dispersing responsibilities and resources among the participating parties [5]. Sharing responsibilities and resources is especially helpful for outdoor recreation agencies and services primarily in a financially challenging circumstance. Partnerships for governmental agencies at all levels in many cases are required or a precondition for public assistance or funding [37]. Many public outdoor education services, like an education or outreach division of a land management agency, mainly rely on partnerships to serve the public due to insufficient agency resources, personnel, and decreased budgets necessary to provide adequate educational and recreational opportunities to users [38]. Along with sharing resources, parks and recreation agencies benefit by showcasing their facilities and outdoor amenities to the community and participating agencies [39]. With increased exposure to outdoor recreation through community partnerships with parks and recreation agencies, community outdoor equipment retailers and outfitters are likely to see increased sales, and recreation nonprofits have higher membership and participation rates [5].

Many outdoor education centers have successfully partnered with other stakeholders to reach out to more participants and serve the community with a broader range of outdoor activities. Schools and youth-focused organizations are typically essential partners with outdoor education centers. Schools active in partnerships with outdoor organizations report many benefits. Many schools, especially in urban areas, cannot offer outdoor recreation programming without the assistance of community partners. Therefore, partnering with community businesses and organizations assists teachers by lessening the financial and teaching burdens associated with the outdoor recreation curriculum [23]. While partnerships present various advantages for participating organizations, they also come with a variety of challenges. Many of these challenges are the result of poorly defined leadership roles, unequal distribution of responsibilities, and a lack of enthusiasm from one or all parties [39]. While partnerships are typically beneficial for participating agencies, organizations often face difficulty in finding the time necessary to initiate and sustain partnerships [38]. A successful collaborative partnership requires higher-level managers to create time and space for staff to begin and engage with partners [40]. In organizations where partnerships are encouraged, management's desire and push to form partnerships may lead to "partnership fatigue" among the parties involved [39]. In recent years,

multiple states have created offices of Outdoor Recreation intending to form and sustain partnerships in the outdoor recreation sector [41]. The results indicated that personnel aims to partner with federal agencies to capitalize on mutually beneficial networks, support agency personnel, provide an ongoing collaborative framework to increase capacity in the long-term, and generate state-level support for decisions on federal lands [42].

## 3. Methods

A qualitative research approach was chosen for this study to provide an insightful understanding of the complexities of the roles and functions of outdoor recreation education centers, especially the partnership efforts within urban communities. In-depth semi-structured interviews were conducted with outdoor education providers from two service locations and various positions for exploring the similarities and uniqueness of partnerships and programs at each outdoor education center in urban communities. The study was approved by the South Dakota State University (SDSU) Institutional Review Board (IRB) to protect human subjects (IRB-1901019-EXM).

### 3.1. Study Location

Two Outdoor Campuses (OCs) in South Dakota were utilized as study sites in this investigation. These two OCs are located in the two most populous cities in South Dakota: Sioux Falls, with a population of approximately 178,000, and Rapid City with a population of 75,000 (Figure 2). The Wildlife Division of South Dakota Game, Fish, and Parks (SDGFP) oversees both campuses for promoting outdoor education and skill learning opportunities in close proximity to the urban areas in the state. The main goal of the OCs is to preserve outdoor opportunities in the state through partnerships and stewardship to connect the state residents and visitors to the great outdoors in South Dakota. Both campuses provide public physical areas and facilities for exploring outdoors supplemented with programs and outreach services for the surrounding communities and statewide to promote outdoor recreation skills (e.g., hunting, fishing, kayaking, etc.). They also offer conservation and stewardship toward the natural environment. All the services and programs at the campuses are provided at no direct cost to participants as the campuses are funded through the sale of hunting and fishing licenses in South Dakota and Pittman–Robertson/Dingell–Johnson funds. Classes at the facilities are led by a staff of full-time naturalists, seasonal interns, and over 150 volunteers.

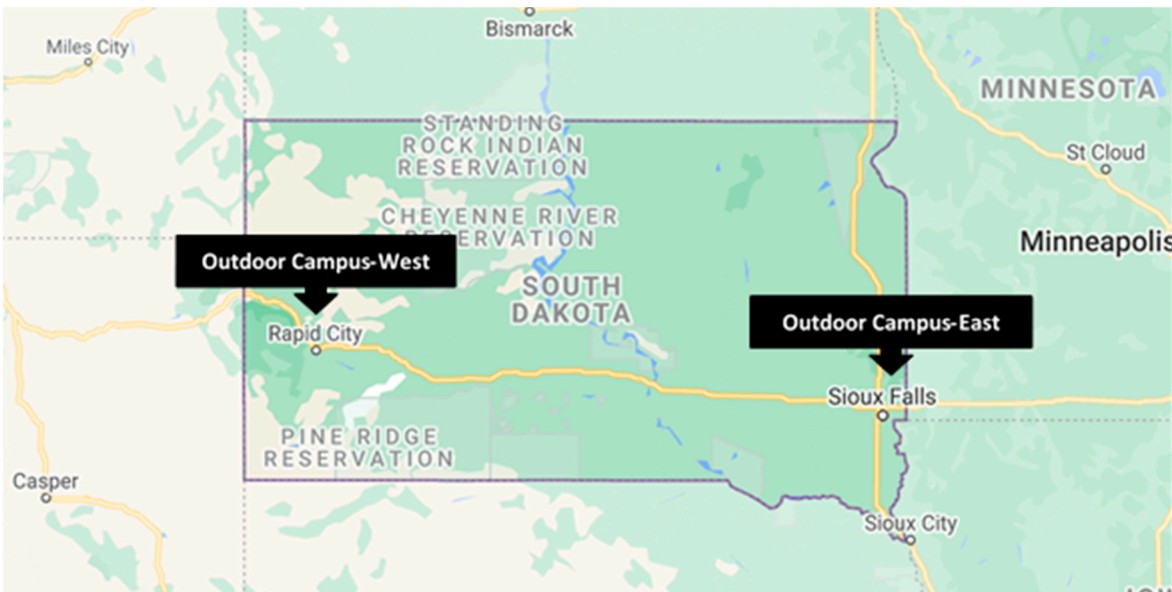

**Figure 2.** Map of outdoor campus locations in South.

In 2019, four main different categories of outdoor programs were offered at both OCs, including environmental education (e.g., wildlife and habitat, lifecycle), outdoor skills (e.g., backpacking, nature photography, outdoor cooking, survival skills), hunting (e.g., BB gun, archery, firearm safety), and fishing (e.g., ice fishing, fly fishing). More than 100 different programs are offered annually: Some are seasonal programs, and some are yearlong programs. All programs are free for youth, adults, and families through an on-site visit, community outreach, and school programs (from K-12). Approximately 40,000 individuals participated in the outdoor programs offered by OCs in 2019.

The Outdoor Campus-East (OC-E) in Sioux Falls, South Dakota, was opened in the summer of 1997. The east campus features over three miles of walking trails located adjacent to one of the city's largest parks, as well as the Sertoma Butterfly House. Fourteen years later, the desire to educate South Dakota's residents and visitors grew, so the Outdoor Campus-West (OC-W) in Rapid City, SD was built to service the west side of the state. The OC-W features an archery park, one-and-one-half miles of hiking trails, and various other amenities. The west campus is unique with its location in the Black Hills of SD. Many of Rapid City's residents are active in outdoor recreation activities in their daily life due to their proximity to state parks, national forests, hiking trails, mountain lakes, and camping areas. The OC-E in Sioux Falls, South, is centrally located within the state's largest city. With fewer traditional outdoor recreation areas in southeastern SD, the OC-E is able to introduce outdoor skills to a wide variety of individuals in the area.

## 3.2. Participants & Data Collection

We applied a purposeful sampling strategy to ensure the inclusion of staff and managers who can provide insightful information and knowledge on the topic. The inclusion criteria are (1) SDGFP employees on both west and east outdoor campus, (2) experience with community engagement and building partnerships to promote outdoor recreation through educational efforts in South Dakota, (3) willingness to voluntarily participate in the project. The original invitation was sent out using participants' email addresses obtained from the SDGFP website. Participation in the study was completely voluntary, and participants could cease involvement at any time. No risks or direct personal benefits were associated with participation in the study.

Seven face-to-face interview sessions were conducted with a total of eight outdoor educators from two service sites in the state of South Dakota. All interviews were conducted between February and May of 2019 in a conference room at the study sites, except for one interview with two interviewees was conducted at a university conference room due to the interviewees' travel schedules. Both settings provided a quiet and comfortable environment for a face-to-face interview. The first and second author conducted all interviews in person and audio recorded; each lasted between 50 to 80 min. Audio files were stored on secured electronic devices and then transcribed verbatim. There were four participants from each campus in the study, including three males and five females between the age of 35 to 64 with 5–25 years of outdoor recreation-related service experience. These eight interviewees were named from participant 1 (P1) to participant 8 (P8) to protect interviewees' identification and confidentiality.

## 3.3. Interview Structure

A semi-structured interview was used in the study, which allowed researchers to ask questions structured for consistency with clear guidelines reflecting the main purpose of the study while also offering the flexibility for research participants to talk about their experience and perspectives. There were three main sections in each interview: (1) participants' background information, such as their roles, responsibilities, and experience working at the campus, (2) various collaborative efforts with the community and partnerships through which they inform, create, and mention other organizations and agencies, and (3) changes, challenges, and opportunities of programs in promoting outdoor recreation to the public and with community partners. The interview questions were created based on Larson et al.'s social structure [18], derived from socio-ecological model for the outdoor recreation-related topic. This approach allowed educators to not only share their personal values, beliefs, and experiences in

outdoor education, but also to emphasize the role and scale of influence of their position and OCs within the community. Also, the authors utilized the socio-ecological model to ask follow-up questions to gain a more in-depth understanding of educators' interaction with the organization and community, their effort to create community networks within the local context, and their decision making process with consideration of organizational policies and environment.

*3.4. Data Analysis*

Qualitative content analysis was applied to identify common themes and essential quotations that emerged from the data analyzed through the interviews. After transcript verification from interviewees, all transcripts were imported into a qualitative computer analysis program (NVivo 12) to improve the data management, coding process, and analyses. The study employed an inductive analysis. First, both authors read and familiarized themselves with all transcribed interviews of eight educators. Next, the two authors began open coding all transcripts individually and identifying patterns from each interview. They also used their interview notes to assist their individual coding process since both authors were involved in data collection/interview. The authors then discussed the connection and linkages among the codes and the themes they identified. To ensure consistency of the coding process, the authors met on a weekly basis seeking to establish consensus regarding emerging themes and to identify similarities and differences from the data. When a significant difference occurred, existing literature and conceptual models were revisited for additional context and insight. Overall, the coding pattern and thematic identification revealed about 80% similarity between the two authors, while some discussion and clarification were needed for emerging concise themes and reaching consensus agreement of results.

## 4. Results

The results of the study revealed three main themes. The first theme, the *gateway to our outdoor legacy*, is more general in both partnership and programming. The second and third themes focused on partnership and programming, respectively.

*4.1. Gateway to Our Outdoor Legacy*

The first emerging theme from these interviews is the pride and commitment to the long-lasting outdoor legacy of the state. This theme is related to the macro-level factors in social-ecological model, such as social and cultural environment as well as political environment in South Dakota. All interviewees commented on the ability of the outdoor education centers and programs offered to cultivate outdoor culture and carry on South Dakota's outdoor legacy. They all understood the importance of outdoor education centers in South Dakota, famous for prolific resources and opportunities for outdoor recreation. They also recognized the uniqueness of the outdoor campus comparing to other outdoor education centers elsewhere. P5 and P8 further explained the essence of the outdoor campus to focus on outdoor skills education, such as hunting, fishing, and other outdoor-related skills, rather than environmental education as traditional outdoor education or nature centers. P8 stated: *"I don't think there are many places that balance environmental education with hunting and fishing recruitment, retention, and reactivation. . . . Most states that do outreach don't have campuses to do outreach with. That's something special about South Dakota."* However, some confusion might exist due to misconceptions of why a nature center (outdoor campuses) offers environmental education along with hunting and fishing instruction. An outdoor campus in South Dakota may not be a typical environmental education center in a different setting. However, in South Dakota, an outdoor skill education center focused on conserving the state's outdoor heritage reflects the values and ethics present in the population.

Moreover, most educators maintained broader influential factors, such as new leadership and priorities in the state government, considerably shifted the focus on what programs are priorities and what evaluation tools are effective to evaluate impacts of OCs on the agency's overall goals and

objectives. For example, fishing and hunting license sales are essential for the agency's operation sustainability; therefore, hunting and fishing-related programs have become priorities which result in reducing other relatively indirect-yet relevant—programs such as arts and crafts, book reading, etc. All educators agreed such change is understandable and beneficial for the agency in the long-term, although some found the change challenging at the personal level. Despite some personal value difference with the organizational priority, all educators recognized their role as public servants is to serve the community and fulfill the agency's mission to "optimize the quantity and quality of sustainable hunting, fishing, camping, trapping and other outdoor recreational opportunities." [43]. Most importantly, they believe that they played a vital role in continuing South Dakota's outdoor legacy in creating an environment with a variety of opportunities for learning about the outdoors and related skills. P7 further explained the recognition of OC and stated:

> "One of the benefits in working in an outdoor campus setting is that you have that recognizable footprint within the community and in the local area so the state, regional, national organizations that are trying to do similar things to what we're doing come to us."

Most interviewees emphasized that the centers provide outdoor campus opportunities for children and youth in the city to explore nature and enjoy the outdoors (e.g., outdoor play, nature playground) and to learn new skills and knowledge in a safe outdoor space (e.g., fishing, shooting bows and arrows); otherwise they have no or limited access to experience. P5 said, *"We're giving people maybe their first time to canoe, kayak, shoot a gun, or do archery. I think we're the one opportunity urban people have to discover outdoor recreation."* Moreover, all interviewees spoke about the effect on children and youth programs they have observed on participants and aiming for greater outdoor involvement and recreation participation. For example, P1 explained " … *teaching outdoor recreation, it influences those kids (participants) to go and influence other kids and the neighbor kids down the road."* These participants can share their new outdoor recreation knowledge with others in their lives and spread the culture of the OC.

All interviewees mentioned the impacts and connection of the outdoor education centers on the individuals and the community, which shows the meso level of influence of OCs within the community. First, the centers help people to build their confidence in the outdoors. Focusing on outdoor skill education, all the interviewees noted the unique outdoor recreation opportunities in SD and the ability to instill confidence in outdoor skills as a common outcome of program participation. Second, the outdoor centers served as a hub to help people new to the areas get to know the culture and environment of the community and the state. P7 expressed the rise in confidence among participants who are new to the area or unfamiliar with the outdoors: *"What do I need to do for going on a hike? It seems to be very basic to us, but people new to the area or have just never been outside, it builds a lot of confidence."* With increased confidence in outdoor skills, interviewees also note the creation of a sense of community. P1 and P4 both noted the ability to provide a safe outdoor space in the city as an important impact of the OC.

*4.2. Working Together for Outdoor Education*

4.2.1. Formal Partnership

Educator interviews revealed three categories of partnerships: local community partners, cross-jurisdiction, and interagency collaboration. Common local community partners consisted of school districts, parks and recreation departments, landowners, nonprofits, and sports outfitters. The Rapid City and Sioux Falls school districts work with their respective OC to contract a teacher to the campuses. The Sioux Falls school district hires a teacher to work full-time at the OC-E, while the Rapid City school district hires a full-time teacher for the OC-W who are responsible for teaching school curriculum for field trips in an outdoor education setting. These teachers are then responsible for planning and instructing outdoor education courses for the cities' schools while fulfilling curriculum needs for schools such as life sciences and social sciences (P2, P4, & P5).

The OC-E also has a formal partnership with the City of Sioux Falls Parks and Recreation Department. The east campus sits on about 100 acres of land within a Sioux Falls city park. The city

parks and recreation department manages the outdoor area while the OC-E manages the building. Within this partnership, the OC-E also assists with fishing, archery, and snowshoeing programs organized by the Sioux Falls Parks and Recreation Department. Besides partnerships with outside organizations, the campuses exhibit a high level of collaboration between two campuses. Regarding the campuses working together, one interviewee stated that *"there is rarely a day that goes by that we don't talk, text, or email one thing or another back and forth. We have a strong connection."* While OCs in two different locations may have different audiences, interviewees still see the need to keep in touch and work together to accomplish the goals of the agency.

### 4.2.2. Programmatic Partnership

For programmatic needs, interviewees indicated that nonprofits and landowners are essential to create new programs as well as reach audiences with similar goals (P4 & P6). Several nonprofits organizations were identified as programmatic partners, such as youth-focused groups, church/religious groups, and volunteers. In recent years, P1 and P7 also have turned to partner with landowners in order to fulfill their first-time hunters' program. The coordinators work with landowners with an abundance of specific wildlife populations. The program accomplishes wildlife management for landowners while instructing participants related to the sport of hunting. One interviewee explained the hunting program saying:

> "I takeout first-time youth and adult hunters, and I take them to the gun range where we shoot and get them comfortable with the gun, and then I actually take them on an actual hunt where they actually harvest and process a deer."

P1 commented that many private landowners enjoy being a part of a person's first hunt, especially youth hunters. After taking a group of hunters out one evening, one landowner told P6 how much he enjoyed their presence and compared it to having his granddaughters home again.

Along with private landowners, interviewees also seek to profit with local nonprofit groups and sporting goods stores. One organization, Trout Unlimited, assists with teaching classes at the OC, bringing their expertise to classes like fly-fishing. In turn, participants from these classes are going on to become members of Trout Unlimited and progressing their conservation efforts. Both OCs also work with sporting goods outfitters in their respective areas. In one example from interviewees, the sporting goods store asked if the OC could give out hats with the store's name as well as participate in their major summer event, Outdoor University. Through this partnership, both agencies are able to promote outdoor sports. Individually, the OC benefits through receiving more supplies while the new sporting goods store benefits by putting their name out in the community.

### 4.2.3. Finding Balance in Partnerships

Interviewees from both campuses noted the need to find a balance in partnerships. Partnerships often originate from acknowledging similar or like goals; however, P6 noted the need to ensure that partnerships do not stray too far away from The OC's ultimate goal and mission. All the interviewees commented regarding their openness and willingness to collaborate with a variety of partners, while recognizing that not all partnerships are perceived as mutually beneficial or equally valuable on a personal level. For example, P1 addressed this need to find the balance in partnership with a local youth service entity and stated:

> "(A non-profit) have kids that pay to come attend their (summer) camp, but they bring them here to do fishing or archery. But it's still bringing people, and allowing us to introduce them (youth) to the outdoors. Even though we are not profiting on it and they're making a profit, it's still a collective audience that they are bringing to us that wouldn't normally be here. We get hundreds of kids that come in through another organization here."

All educators recognized partnerships require time and invested energy to reach specific goals. Building partnerships is a dynamic process; maintaining partnerships require time and effort.

Partnerships need time to grow and mature; most partnerships with non-profit or other public agencies show steady growth or maintenance; however, partnership with private sector agencies show less stability. P5 explained a frustrating experience with a cooperating retail store with which they had a mutually beneficial partnership for years. That cooperation ended due to a top-down decision from the headquarters. Yet, a new opportunity came when a new outdoor retail store opened in town, becoming the major sponsor for events and offering great deals for outdoor equipment and promotional materials.

Outdoor educators must be creative in coming up with innovative strategies to provide programming with other organizations that align with the mission of the OC and the partner organization's mission and goals. P6 used an example to explain the partnership and said:

> "We know that some of the people involved with a homegrown group are interested in growing their own food from farm to table and field to table. Well, if you're harvesting your own food and you're raising your own chickens, maybe then you'll go hunting. There's a similar connection there. I call them gateway classes. We have a common interest, so let's see if we can cross over a little bit."

To maintain these partnerships, P2 emphasized the need for effective and frequent communication between partners to ensure that both organizations are benefitting from the collaboration. P6 described the approach to open communication and evaluating the success of a partnership, such as "am I giving you what you need?" and "have I held up my end of the deal?" while simultaneously informing the partner organization how they have been doing.

*4.3. Challenges and Opportunities in Programming*

4.3.1. Common Challenges in Outdoor Education Programs

The abundance of programming offered at the OCs is accompanied by challenges such as limited resources, participant accountability, and marketing. Interviewees from both campuses noted limited resources like space, staff, and funding as programming constraints. Attracting and recruiting new participants was also a common challenge on both campuses. With the new focus on recruiting hunters and anglers, the OCs must find new and creative ways to recruit non-consumptive outdoor recreation users to hunting and fishing. The campuses also have the task of attracting non-users to outdoor recreation. P8 addressed such challenges: *"When you work for Game, Fish, and Parks, you engage with hunters and anglers all the time. What's challenging is you don't engage with the non-users. Finding those people means that you have to put yourself into a different position than you normally do."*

While in an urban setting, P5 commented on the challenge of attracting urban participants: *"Urban people don't always think about the outdoors first. . . . They don't always think about a walk through the forest, snowshoeing, or skiing or learning about what's under the water. It's not at the top of mind awareness."* Expanding on the challenge of marketing to new audiences, P8 said *"In order to engage with new audiences, you have to go where those people are and meet them. . . . If we want to engage older populations, we have to put ourselves in places we've never been before, and that's been challenging."*

Participant accountability was also cited as a common challenge among programs. As for free programs, educators noted that participants do not have anything holding them accountable for attending the class. Adult participants are frequent no-shows to classes. By registering for classes and not showing up, potential other participants are unable to attend classes at the OC.

4.3.2. Evolving Process of Outdoor Programs

The transition from youth to family and adult programs was common in every interview. This shift of focus is a result of R3 programming initiatives to recruit, retain, reactivate hunters, and anglers in the state [7]. South Dakota Game, Fish, and Parks has seen a decrease in hunting and fishing license sales over recent years. Along with this transition comes the need to create innovative programs. P5

explained the need to follow the trends as a method to attract new participants to programs. Another creative strategy used to attract more adults was the creation of date night classes. P6 spoke about the difference in attendance due to marketing changes. In the past, the facility struggled to fill adult participant rosters. However, once date nights were advertised, the OC saw an influx of interest in adult programming. In line with the transition to adult and family programming, the facilities are also turning to series classes in order to teach more advanced skills. P6 spoke about the changes to program delivery:

> "We tried to provide every opportunity and round them out as much as possible. ... When it comes that they're (participants) not just taking stuff from us, they're using their resources. But when they come back and give back, it's awesome! That's the goal."

The goal of the new focus on family programs is that outdoor skills become commonplace in families to participate together rather than only educating youth in the activities. P5 noted the benefits of family programming as not just providing youth a single experience but allowing them to participate in outdoor recreation and practice their new skills together as a family activity.

Finally, to engage new or nontraditional hunting and fishing groups, it is crucial to identify these "untouched but with potentials" and reach out to them using the approaches that this target population would accept. Several niche groups or nontraditional outdoor recreation participants were discussed in the study. For example, older adults (50 years and older), and especially women, have become new participants at OCs, so special programs or collaborations with local businesses (e.g., hotel, winery) for specialized packages has been offered to expand their service target group, which might be also beneficial area tourism as well. Also, homeschool programs are the fastest-growing programs on both campuses. P1 explained such discovery with the homeschool group:

> "We started about five or six years ago now. I was at a conference, and they were talking about how the need is out there for home school how they are a collective audience. ... There is a huge, huge following of home schools and they are all looking to tie into something that they can actually ... away from home with their peers and stuff like that too so there has been a huge following. ... Their flexibility is as far as schedules is a lot easier as well. Plus, the home school community is sometimes a little more open maybe to different ideas and stuff like that as well."

In order to increase the mobility of reaching out to communities across the state, outreach programs play essential roles in interacting with community members. It provides direct services to those who might not be able to travel to the campus for learning shooting, fishing, and archery. Also, some efforts have been made to have outreach to college students through training and workshops in broadening the scope of the services.

Finally, another prevalent theme in the interviews is the change in evaluation, which is usually considered as the last phase of programming in which values and impacts of services need to be assessed and used for the improvement of the next program cycle. Although some evaluation techniques or tools have been utilized on both campuses for program evaluation, it has been challenging to demonstrate the impacts of outdoor education programs without continuous tracking and consistent documentation. Several interviewees addressed such essential changes in the outdoor education system and hope the new data tracking system is able to produce reliable data and provide a stateside outdoor education effort and impacts for informed decision making in operation, management, investment, and budgeting for future outdoor education.

## 5. Discussion

The purpose of the study is to understand the challenges and opportunities of urban outdoor education centers in applying innovative programs to research out to the public and utilizing partnerships to create a social network in the community for enhancing the culture of outdoor

recreation and environmental conservation. Two outdoor education centers, Outdoor Campus East and West, in SD urban areas, were used to understand how outdoor education centers as a gateway entity serving as the center of building community support and networks for outdoor education and promoting outdoor recreation in urban areas (Figure 3).

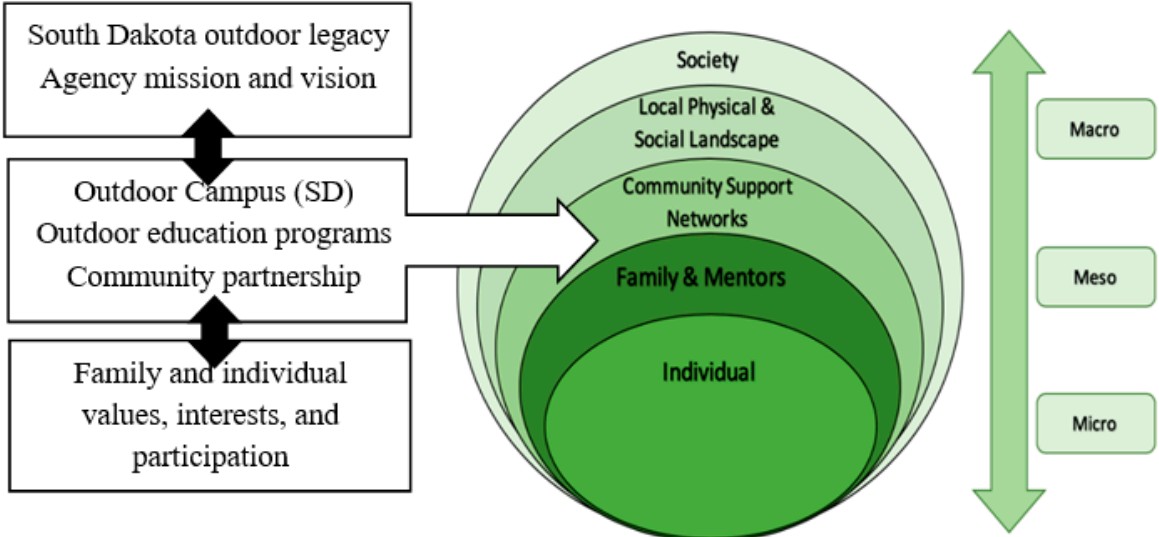

**Figure 3.** Outdoor Campus impacts and relationships in social structure for outdoor education.

First of all, the results indicated that outdoor educators play essential roles to support the foundation of OCs by creating an environment supportive of individuals pursuing outdoor recreation or organizations with compatible goals and interests in cultivating outdoor cultures. The OCs provide opportunities and enhance accessibilities through various approaches (e.g., on-site programs, outreach,) and collaborations (e.g., formal partnership, programmatic partnership). As a state entity, the OCs tended to work with a wide range of partners to reach out to people and organizations with different interests and roles in outdoor recreation and education. It is the reason why the educators expressed the importance of finding balance in partnership, so they can sustain the partnership without losing the agency's overall mission and vision while fulfilling the community's needs and interests. It is important to notice that as social contexts shift, such as modernization and urbanization, individuals' values and beliefs of outdoor education related to wildlife conservation also change. At the cultural level, residents in urban areas typically experience higher levels of modernization, typically show higher education levels, frequently have higher incomes, and tend to view wildlife as part of an ecological network worthy of care (mutualism), rather than viewing wildlife as benefiting humans through hunting or fishing (domination value) [44]. Therefore, wildlife management agencies might need to incorporate both domination values and mutualism values in decision-making processes for reducing the potential conflicts in the perception of wildlife conservation, especially in urban areas.

Moreover, the results also showed that the OCs could be viewed as meso-level factors in the social–ecological model to serve as a hub providing outdoor education and skills programs and facilitating the community for creating networks, interactions, and connections between organizations directly or indirectly associated with outdoor recreation. Urban outdoor education centers are essential influences in the promotion of statewide outdoor recreation opportunities and resources and in creating and maintaining partnerships with public, private and non-profits sectors for serving people with a wide range of interests and skill levels in outdoor recreation. Using social–ecological framework, the OCs in urban areas only create a social habitat to sustain the environment for outdoor recreation but also foster the interaction among various organizations to provide a supportive social network, creating a sense of community through engaging outdoors and natural environment [18,45].

Another interesting finding was the dynamic and evolving process of outdoor education programming revealed in the study. All outdoor educators recognized that the SD outdoor education system is a unique model in the nation. The OCs are not typical nature centers or environmental education units. Both environmental education and outdoor skills are part of the scope of outdoor education [17]. The primary focus of environmental education is to explore environmental issues, engage in problem-solving, and take action to improve the environment [46]. The OCs provide both environmental education and outdoor skills, which might create some confusion to the public with conflicting ideas of protecting resources (environmental education) and consuming resources (outdoor skills, such as hunting and trapping). It has been an evolving and learning process for many research participants. It takes time, effective communication, and teamwork to find a balance between how to provide both environmental education and outdoor skills, how many programs of each category should be offered, and what programs are needed and appropriate for each location. With the current shift from youth-focused programs to family and adult-oriented shooting sport, although some learning curves were discussed, all research participants understood the reason for changes and recognized the benefits and impacts of reaching out to various untouched populations or untraditional outdoor participants and hunters and anglers (e.g., older adults, minorities, women). As there have been declining fishing and hunting license sales in SD, the OCs could be viewed as an education and outreach center to recruit, retain, reactivate (3R) hunters and anglers in the state [7].

Finally, this study has several research limitations that render opportunities for future research. First, the study only included two outdoor education centers in South Dakota as study sites and recruited eight individual educators for the study. It is possible that the response is location-specific with place-based experiences and practices which might not necessarily be appropriate for other outdoor education centers. Semi-structured interviews from the OCs educators provided insightful knowledge and information regarding their experience and responsibilities in promoting outdoor recreation within a wildlife- and natural resource-related governmental agency through educational efforts. Although we were unable to interview all full-time outdoor educators from OCs, these eight volunteer educators participating in the study represent approximately one-half of the full-time employees from both outdoor campuses with a wide range of expertise and experience within SDGFP. At the state level, it might be helpful to conduct interviews using similar questions and techniques with other outdoor education centers at federal (e.g., National Park Service) and local (e.g., municipality) levels to explore the cross-agency and jurisdiction partnership and collaboration. Moreover, it might be beneficial to identify other education centers or organizations with similar priority in promoting fishing and hunting for conducting thorough case studies with interviews to explore the challenges and solutions in such topics.

**Author Contributions:** Conceptualization, P.O. and H.-L.L.; Methods, P.O. and H.-L.L.; software and formal analysis P.O. and H.-L.L.; writing—original draft preparation, P.O.;writing—review and editing, H.-L.L. All authors have read and agreed to the published version of the manuscript.

**Funding:** This work was supported by the USDA National Institute of Food and Agriculture (NIFA), Hatch project (#1016822).

**Acknowledgments:** We greatly appreciate the collebration of Outdoor Campus-East and Outdoor Campus-West and the outdoor educators who volunterred to share their experiecne and knowledge with us for this study. We also like to thank Lowell Caneday and Kiley B. Foss for their valuable and helpful edits and suggestions to this mannusipt.

**Conflicts of Interest:** The authors declare no conflict of interest.

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
