# Peer review of "Gateway to Outdoors: Partnership and Programming of Outdoor Education Centers in Urban Areas"

_education, doi:10.3390/educsci10110340_

Round 1

Reviewer 1 Report

Dear colleagues,

The topic you chose for the article is of high interest due to the attention paid nowadays to extracurricular and outdoor activities.

Yet there are some questions remaining:

  1. Do you think it would add a bit more value for your research if the map of the outdoor campuses under analysis were attached?
  2. Will you, please, elaborate on the aforesaid campuses providing:
    • the information about the capacity (day or week or month...)
    • samples of the structured programmes (curricula) that are already applied in the activities
  3. Are you absolutely confident that eight interviewees only make up a representative sampling? Isn't it reasonable to increase the number to make the research much more scientifically sound and based?
  4. Will you consider to add diagrams and tables summarising the findings to make the results more comprehensible?

I believe it won't take you much time and effort to put these final touches to your work to add an extra value to it.

P.S. On page 14 there is a misprint: "takeout" instead of "take out"

Sincerely yours

Author Response

Dear Reviewer:

Thank you for providing constructive feedback for us to revise the manuscript. We incorporated all the suggestions and recommendations Please see the detailed response and location for major modifications in the track change document. We look forward to hearing from you about our revision.

Based on your suggestions, we made the following changes:

  1. Included a map on p. 12, line 248
  2. Provided more details in OCs capacity and programs. P.11, line 228-235
  3. Addressed the sampling and representation of the research participants in discussion. P. 27/28, line 606-609
  4. Added a diagram in discussion. P. 26. line 548
  5. Made editing suggestions and overall editing effort

Reviewer 2 Report

Potentially there are some interesting aspects to this paper, but I think it needs a very serious revision.  I thought that it relied on many very general research findings to establish the rationale of the study. It also relied on the an assumption that these are 'traditional experiences' and therefore beneficial.  It would be helpful to analyse this asumption.

There was a lot that wasn’t explained about the context.  For example, what is an outdoor heritage activity and why should these activities be preserved?  Are these traditional colonial activities?  What type of hunting would occur?  Hunting and fishing and even the continued presence of groups of people can disrupt natural environments, but this doesn’t seem to be addressed as an issue.  I was concerned that there might be implicit elements of settler colonialism within the approach (see e.g. Bacon, J. M. (2019). Settler colonialism as eco-social structure and the production of colonial ecological violence. Environmental Sociology, 5(1), 59-69.).  It is important that this is analysed because, without analysis, this could be seen as practices of traditional owners, but when analysed it becomes much more problematic. 

There is a comment about landowners having an abundance of specific wild animals (p.14) – I wondered how that decision was made and if it is made from a domination type approach (see Manfredo, M. J., Teel, T. L., Don Carlos, A. W., Sullivan, L., Bright, A. D., Dietsch, A. M., ... & Fulton, D. (2020). The changing sociocultural context of wildlife conservation. Conservation Biology.)  It also seemed that some children are coming for very specific experiences (e.g. the non-profit mentioned on p.16) and missing the broader environmental education or even potentially the skill acquisition.  I’m not sure how these work together.  It seemed to be the case that they are happy to take any partners.

On a related point, the conflicts of interest in funding were mentioned but not analysed.  The profits mentioned on p.16 are for partners who seem selective about activities.  The overall program is funded through the sales of hunting and fishing licenses and the experience provides training in hunting and fishing. These are fundamental partner issues that are likely to influence the activities that occur.

The interview structure was: “(1) participants’ background information, such as their roles, responsibilities, and experience working at the campus, (2) various collaborative efforts with the community and partnerships through which they inform, create, and mention other organizations and agencies, and (3) changes, challenges, and opportunities of programs in promoting outdoor recreation to the public and with community partners.”  It would help to make clear how the interview related to the social-ecological framework used, particularly the claim on p.6 that “individuals’ outdoor recreation participation is influenced by higher-social order, environmental, and policy-related structures”  The social-ecological framework could also be used more clearly in the discussion.

More detail is required about the thematic analysis conducted.  Was there a published process followed or did the researchers develop their own process?  Were there any disagreements?  If so, how were these resolved.  I was also unclear on how the themes were identified.  I’m guessing it was inductive as the first sentence on p.11 states “Qualitative content analysis was applied to identify common themes and essential quotations that emerged from the data analyzed through the interviews.” – but if it was inductive, how did both coders arrive at the same themes?  More detail is needed.  I’m also unsure why the existing literature and conceptual models were used for triangulation.  It would be more typical to use documentation related to the topic of investigation e.g. reports from the OCs.

I have concerns that citations have sometimes captured what might be convenient for the authors arguments but misrepresent the content within the articles cited.  For example, page 2 “Especially in urban areas, outdoor education centers not only provide opportunities for recreational involvement and allow personal growth and learning in a unique setting but they also promote ecosystem health and human well-being in the community [8,9].” – Reference 8 does not include findings regarding ecosystem health.  Reference 9 doesn’t make any claims about outdoor education centres. Outdoor education in reference 9 is sometimes included as an example of use of green space, but there isn’t a clear finding about outdoor education and ecosystem health.

There is clearly a popularity of the OCs and an importance to many people in South Dakota along with a motivation to maintain and expand the read of the education. This focus of the manuscript could potentially be enhanced.

Author Response

Dear Reviewer:

Thank you for providing constructive feedback for us to revise the manuscript. You have provided many useful comments for us to revise the manuscript. We incorporated all the suggestions and recommendations Please see the detailed response and location for major modifications in the track change document. We look forward to hearing from you about our revision.

Based on your review, we have (1) bulked up and improved the introduction to the paper especially in the context of South Dakota, (2) added some suggested literature where appropriate; (3) provided more details in methods (e.g., analysis, interview, etc.), (4) linked the results with the model in the discussion and (5) additional editing efforts.

Thank you

Round 2

Reviewer 2 Report

I've read through the response from the authors and the revised manuscript.  I appreciate the additional information lines 45-63.  It provided a good context for the paper.  I am glad the authors were able to accommodate all of the revisions.  I could also see from this version that the authors have made clear the broader range of influences, such as historical and attitudinal changes that will continue to be a challenge for the OCs.

It was a pleasure to review the revised version.